

# Stock trend prediction using sentiment analysis

Qianyi Xiao and Baha Ihnaini

Department of Computer Science, Wenzhou Kean University, Wenzhou, Zhejiang, China

## ABSTRACT

These days, the vast amount of data generated on the Internet is a new treasure trove for investors. They can utilize text mining and sentiment analysis techniques to reflect investors' confidence in specific stocks in order to make the most accurate decision. Most previous research just sums up the text sentiment score on each natural day and uses such aggregated score to predict various stock trends. However, the natural day aggregated score may not be useful in predicting different stock trends. Therefore, in this research, we designed two different time divisions: $0:00_t\sim0:00_{t+1}$ and $9:30_t\sim9:30_{t+1}$ to study how tweets and news from the different periods can predict the next-day stock trend. 260,000 tweets and 6,000 news from Service stocks (Amazon, Netflix) and Technology stocks (Apple, Microsoft) were selected to conduct the research. The experimental result shows that opening hours division ($9:30_t\sim9:30_{t+1}$) outperformed natural hours division ($0:00_t\sim0:00_{t+1}$).

## INTRODUCTION

For decades, stock trend prediction has been a popular topic due to its importance in the economy and risk management (*Ou & Penman, 1989*; *Dimson, 1979*; *Checkley, Higón & Alles, 2017*; *Ranaldi, Gerardi & Fallucchi, 2022*). However, its inborn complexity and uncertainty decide its difficulty (*Agrawal, Chourasia & Mittra, 2013*; *Holthausen & Larcker, 1992*). For example, politics, wars, and many factors would sharply affect stock prices (*Agrawal, Chourasia & Mittra, 2013*; *Alzazah & Cheng, 2020*). Thus, achieving the best result with the minimum required data is the goal (*Agrawal, Chourasia & Mittra, 2013*). In the past, most research employed many financial or technical factors, which aimed to reflect investors' interest and predict stock from a financial perspective (*Weng, Ahmed & Megahed, 2017*; *Bujari, Furini & Laina, 2017*), while a few years ago, with the rapid expansion of social media, people could easily post and spread their emotions through micro-blogging (*e.g.*, Twitter and Reddit) (*Checkley, Higón & Alles, 2017*; *Lassen & Brown, 2011*). Therefore, researchers attempted to utilize micro-blogging data to directly attain investors' moods regarding the stock market, as financial decisions are significantly driven by emotion and mood (*Bujari, Furini & Laina, 2017*; *Nofsinger, 2005*). This provides the theoretical foundation for connecting social media sentiments and stock fluctuations.

Corresponding author
Baha Ihnaini, bihnaini@kean.edu

Twitter, one of the micro-blogging platforms, is a significant data source due to its popularity, transparency, and timeliness (*Checkley, Higón & Alles, 2017*). Twitter provides a cash-tag symbol ($) search to obtain relevant stock Twitter messages (tweets) (*Mudinas, Zhang & Levene, 2019*). For example, $AAPL is the stock topic only for Apple Inc. Therefore, by harvesting tweets with $AAPL, we can build one dataset containing all AAPL stock discussions.

In addition, news is also considered significant in stock prediction and has been widely used for many years (*Kabbani & Usta, 2022*). An expert's authoritative opinions, the description of one company's business and many other factors hidden in the news can motivate investors to buy or sell stocks (*García-Méndez, De Arriba Pérez & González-Castaño, 2022*; *Hao & Chen-Burger, 2022*; *Li & Pan, 2022*; *Sharma & Bhalla, 2022*; *Srivastava, Tiwari & Gupta, 2022*). Among various financial media, Bloomberg, Forbes, and Reuters are some valuable financial news sources (*García-Méndez, De Arriba Pérez & González-Castaño, 2022*; *Li & Pan, 2022*; *Srivastava, Tiwari & Gupta, 2022*). In our research, we will use news attained from eight popular and reputable websites or medias to conduct the stock trend prediction.

Natural language processing (NLP) is one computational technique to analyze and understand human language (*Cambria & White, 2014*; *Hirschberg & Manning, 2015*). Sentiment analysis, a branch of the NLP, identifies and extracts people's subjective attitudes, opinions, and emotions (*Khedr & Yaseen, 2017*; *Hussein, 2018*; *Chalothom & Ellman, 2015*). Sentiment analysis has been widely used in checking online comments, and the main goal is to examine sentiment scores (*Hussein, 2018*; *Chalothom & Ellman, 2015*). This research will adopt sentiment analysis technology as the primary text analysis method.

In this article, we harvest 2021 tweets and news mentioned Service stock AMZN, NFLX, and Technology Stock AAPL, MSFT. We combine VADER sentiment analysis along with other extracted tweets features to generate a novel weighted sentiment index $T_{weighted}$ for each tweet. We utilize FinBERT to analyze news titles and generate $N_{weighted}$ to represent news opinion value. In addition, we also design two time divisions: $\mathbf{0:00_t \sim 0:00_{t+1}}$ and $\mathbf{9:30_t \sim 9:30_{t+1}}$ to explore how tweets and news from different periods can predict the future stock trend. *Goal I: $Open_{t+1}$ - $Open_t$* and *Goal II: $Open_{t+1}$ - $Close_t$* (t is given transaction date) are created to evaluate their performance separately. Finally, six classifier algorithms (KNN, Tree, SVM, Random Forest, Naïve Bayes, Logistic Regression) are applied to check the result with 10-fold cross-validation.

## Related work

In the work of *Bujari, Furini & Laina (2017)*, the authors only used tweets features and achieved 82% accuracy with an *ad-hoc* model. However, no unified model fits all cases, and the prediction accuracy varies greatly for different stocks with the same model. The limitation of this research is the considerably short period, only 70 days (including non-transaction days) of data were collected. In *Checkley, Higón & Alles (2017)*, the dataset goes from the 17th February 2012 to 17th October 2014. The authors proposed a model to predict market volatility, volume, and returns (direction) by using data granular to two-minute intervals. The evidence showed a causal link between all three target and bull-bearish sentiment metrics. Among the five stocks investigated, market volatility and

volume are more predictable than market returns. A similar approach using more granular data also appeared in *Kinyua et al. (2021)*, and they analyzed the impact of President Trump's tweets on SPX and INDU in a 30-minute event window (15 min before tweet posted and 15 min after tweet posted). To focus on the immediate influence of Trump's tweets, only tweets in transaction hours were kept in this research. The model used random forest, decision tree, and logistic regression algorithm to predict how market response to Trump's tweets. Machine learning algorithms showed that the inclusion of Trump's tweets significantly decreased prediction RMSE. Both Trump's strong negative and positive sentiments resulted in an uptrend of SPX and INDU indices. While, for all other sentiment categories, Trump's tweets caused a downtrend.

In addition, *Maqsood et al. (2020)* built a stock trend prediction model based on tweets related to local or global events. The significant events from 2012 to 2016 in USA, Hong Kong, Turkey, and Pakistan markets were investigated. For example, in USA market, the author examined how tweets mentioned the US election 2012 (local event) and Brexit 2016 (global event) affected Apple and Google stock fluctuations. The percentage of positive and negative tweets was calculated per day for each event. The result revealed that not all major events severely impact the stock market. However, more important events like the US election can strongly affect algorithm performance.

Many researchers have tried to make predictions by using data other than tweets. *Kabbani & Usta (2022)* used financial articles from 2016-01-01 to 2020-04-01 to predict the stock trend inside a given day. Considering the rapid change of sentiments, only today and tomorrow trend is predicted. After correlation analysis, highly correlated features with the article's sentiment scores were selected in the final data set. The model used linear regression, random forest and the Gradient Boosting Machine Algorithm, and hit 63.58% accuracy on average. *Weng, Ahmed & Megahed (2017)* introduced a model to predict the stock movement one day ahead. Different from sentiment analysis, the author used Wikipedia traffic, Google news counts, some market data, and various technical indicators to make predictions. The model just focused on Apple stock fluctuations from May 1st, 2012, to June 1st, 2015. Combining data from multiple sources, their expert system hit 85.8% accuracy, reflecting that the increase in data categories can boost the prediction result.

*Mudinas, Zhang & Levene (2019)* divided news and tweets sentiments into eight emotions (*e.g.*, anger, fear). Only a few sentiment emotions showed some correlation with the future stock movements. In most cases, the prediction based on technical factors achieves a better result without emotions analysis. The same conclusion also got in *Khedr & Yaseen (2017)*. They used N-gram and TF-IDF method to generate the weight for each token and classify collected financial news into positive and negative sentiment attributes. The sentiment attributes method achieved 59.18% accuracy in the K-NN classifier, and sentiment and historical stock data method achieved 89.80% accuracy. However, the author did not show the accuracy achieved with sole historical data. It seems that technical factors are the determining factors in stock prediction while extract sentiment attributes have harmful effects on the result, similar to the conclusion of *Mudinas, Zhang & Levene (2019)*.

Except for traditional Machine learning methods, new technologies were also applied in stock predictions. *Nguyen, Shirai & Velcin (2015)* designed a novel aspect-based sentiment

method to calculate the sentimental values for each topic in one sentence. Instead of extracting hidden topics together with sentiments, this new model represents each message as a set of topics with corresponding sentimental values. To examine the efficiency of the novel aspect-based sentiment method, the authors applied SVM on 18 stocks data from July 2012 to July 2013. The aspect-based sentiment feature method achieved 71.05% high accuracy for Amazon stock. *Qiu, Song & Chen (2022)* created a new sentiment analysis model based on Baidu AI Cloud, which provides an algorithm automatically mines sentiment knowledge through unsupervised learning. Also, a novel weighted sentiment index considers the holiday effect. Sentiments in non-transaction were somehow allocated to the next transaction day. This novel index increased seven of the eight algorithms' performance compared to the sentiment index without the holiday effect. Cryptocurrency price forecasting is also a hot topic outside of stock prediction. *Parekh et al. (2022)* presented a hybrid model, *DL-GuesS* in cryptocurrency price prediction considering historical price and Twitter sentiments. *DL-GuesS* model adopted LSTM and GRU deep learning to predict cryptocurrency prices considering different window sizes. The proposed *DL-GuesS* model achieved better Bitcoin-Cash performance than the model only using Bitcoin-Cash price.

## MATERIALS & METHODS

The proposed model presented in Fig. 1 is designed to predict stock fluctuations and help stock investors make proper investment decisions. For news and tweets data, they are first sent to the sentiment analysis test section to select the most appropriate technology for applying sentiment analysis respectively. Then both generated weighted scores will be summed up in given intervals and apply holiday effect processing. Finally, summed news and tweets scores are combined according to intervals as input features. While for stock data, we turn stock fluctuations into binary changes and regard such change as the expected result in machine learning.

### Data collection

**Tweets:** Compared with Twitter official API, the Twint library has many advantages and has been used in research (*Dutta et al., 2021*; *Bruno Taborda et al., 2021*; *Yohapriyaa & Uma, 2022*). Twint has no restrictions in the official Twitter API and provides many options to filter data (*e.g.*, language selection). We use the cashtag symbol to harvest twitter messages of Service stock AMZN, NFLX and Technology Stock AAPL, MSFT in 2021.

**News:** To avoid the bias of specific financial media, we scrape news from eight websites or medias including CNBC, Forbes, The Street, Reuters, The Motley Fool, Business Insider and Wall Street Journal, Bloomberg (*García-Méndez, De Arriba Pérez & González-Castaño, 2022*; *Li & Pan, 2022*; *Srivastava, Tiwari & Gupta, 2022*). Finally, six thousand pieces of news are retained for this study.

**Stock Data:** Yahoo Finance is an influential financial news and data website widely adopted in stock research (*Ahmar & Del Val, 2020*; *Budiharto, 2021*; *Topcu & Gulal, 2020*). It provides historical stock price fluctuations (*Bujari, Furini & Laina, 2017*). Daily price data has six main features: Open price, Close price (or Adjusted Close price), Highest price,

**Peer**J Computer Science

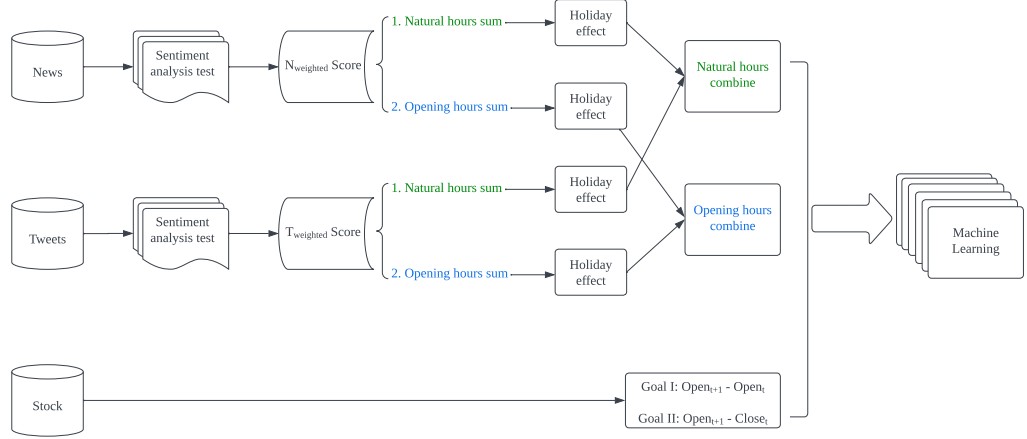

**Figure 1** Flowchart of methodology.

Lowest price, and Volume. We download data for four selected stocks from January 4, 2021 to December 31, 2021.

## Data preprocessing

Because the tweets dataset is built by using text scraped from Twitter, raw tweets with many undesired stuff need to be cleaned. Emoticons, symbols, URLs, and some meaningless texts (stopwords) are removed (*Bruno Taborda et al., 2021*; *Budiharto, 2021*). In addition, tweets containing more than three cashtags are discarded as meaningless messages, the same as Bujari's research (*Bujari, Furini & Laina, 2017*). For news preprocessing, we only keep news that mentioned chosen stock name exactly in titles so that we can avoid the disturb of much unrelated information. Moreover, we calculate the value *Goal I: Open$_{t+1}$ - Open$_t$* and *Goal II: Open$_{t+1}$ - Close$_t$* (t is given transaction date) for each stock, then transform all values into a binary variable, where positive values → 1 (uptrend) and negative values → 0 (downtrend) (*Weng, Ahmed & Megahed, 2017*; *Bujari, Furini & Laina, 2017*). The equation is shown below:

$$Goal\ I = \begin{cases} 1, Open_t \leq Open_{t+1} \\ 0, Open_t > Open_{t+1} \end{cases} . \tag{1}$$

$$Goal\ II = \begin{cases} 1, Close_t \leq Open_{t+1} \\ 0, Close_t > Open_{t+1} \end{cases} . \tag{2}$$

## Sentiment analysis Test
### *Testing datasets to find the most proper sentiment analysis technology*

**Financial PhraseBank:** Financial PhraseBank is one primary dataset for financial area sentiment analysis (*Ding et al., 2022*; *Ye, Lin & Ren, 2021*), which was created by *Malo et al. (2014)*. Financial PhraseBank contains 4,845 news sentences found on the LexisNexis

database and marked by 16 people with finance backgrounds. Annotators were required to label the sentence as positive, negative, or neutral market sentiment (*Malo et al., 2014*). All 4845 sentences were kept with higher than 50% agreement. In our study, we will use this dataset to test the performance of the lexicon or model on news text.

**Tweets_labelled_09042020_16072020:** It consists of 5,000 tweets randomly selected out from 943,672 raw tweets (*Bruno Taborda et al., 2021*). The raw tweets were harvested by using Twitter hashtags and cashtags like #SP500, $MSFT, $AAPL. Inside this file, 1,300 were manually labeled with positive, negative, or neutral market sentiment and reviewed by a second independent annotator. In our study, we will use this dataset to test the performance of the lexicon or model on tweets.

### Sentiment analysis technologies

*VADER sentiment analysis* In our model, VADER library is the first technology used to analyze text sentiment (*Hutto, 2016*). The VADER library is a lexicon and rule-based framework that can detect the three dimensions of sentiment in the text (*Hutto, 2016*; *Pano & Kashef, 2020*). The generated result consists of positive, neutral, negative, and compound scores ranging from $-1$ to 1 for each tweet.

Here, we just apply VADER sentiment analysis and consider the compound score to represent the general sentiment of each tweet. Then, to expand the influence of those popular tweets, we create a new feature Tweets Weighted: $T_{weighted}$

$$T_{weighted} = compound\ score * (retweets + 1) \tag{3}$$

where $T_{weighted}$ is the weighted sentiment compound score for each tweet.

*Loughran-McDonald word dictionary* The Loughran-McDonald dictionary was widely adopted in financial area research (*Wang, Yuan & Wang, 2020*), since it was the first dictionary to explore the potential benefits by using specific financial domain words (*Karalevicius, Degrande & De Weerdt, 2018*). The dictionary was initially developed using corporate 10-k reports between 1994 and 2008 (*Loughran & McDonald, 2011*). We use version 2012, which contains 2,349 negative and 354 positive words to help us evaluate text sentiment (*Loughran & McDonald, 2016*).

*FinBERT Model* FinBERT in a language classification model based on BERT to tackle NLP tasks in the financial domain (*Araci, 2019*). FinBERT was trained by 46,143 news documents TRC2-financial corpus. According to *Nguyen, Shirai & Velcin (2015)*, we generate a News opinion value, News Weighted: $N_{weighted}$

$$N_{weighted} = \frac{pos_{score} - neg_{score}}{pos_{score} + neg_{score}}. \tag{4}$$

## Sentiment analysis test result

Figure 2 shows the testing results of VADER, Loughran-McDonald dictionary and FinBERT performance on dataset Tweets_labelled_09042020_16072020. VADER gets the best classification accuracy 68%; Loughran-McDonald dictionary and FinBERT gain 56%, 53%

classification accuracy respectively. The results reflect that VADER has better classification performance on messy Twitter messages, while the FinBERT trained on well-structured news text gives the worst result in tweet classification. In addition, we can find that VADER far outperforms other two methods in positive tweets classification and VADER also gives twice as many positive labels as the others. VADER tends to label more tweets in the testing dataset as positive than the Loughran-McDonald dictionary and FinBERT do.

Figure 3 contains the classification accuracy of VADER, Loughran-McDonald dictionary and FinBERT on Financial PhraseBank, the result for FinBERT is gathered from *Araci (2019)*, while other two confusion matrix are calculated by using our own results. In news classification, the situation is reversed, FinBERT gets 86% accuracy, but VADER is only 54%. FinBERT is highly adaptive to standard financial text written by professionals compared with tweets proposed by laymen. However, VADER is more suitable for general messages and cannot identify domain-specific terms well.

According to the test results, we will apply VADER analysis on tweets dataset, and FinBERT model on news data we gathered for prediction since they present better sentiment classification results.

## Data manipulation

Because our idea is to explore how tweets and news from different periods can predict the future stock trend, we also design two time divisions: Natural hours division $0:00_t \sim 0:00_{t+1}$ and Opening hours division $9:30_t \sim 9:30_{t+1}$ (t is given natural date) and get Tweets in Natural hours: $TN_t$, and Tweets in Opening hours: $TO_t$

$$TN_t = \sum_{i=0:00_t}^{0:00_{t+1}} T_{weighted} \tag{5}$$

$$TO_t = \sum_{i=9:30_t}^{9:30_{t+1}} T_{weighted}. \tag{6}$$

News in Natural hours: $NN_t$, and News in Opening hours: $NO_t$ are similar to above formular above, except that $T_{weighted}$ is changed to $N_{weighted}$.

The new variables are the summation of sentiment values in each time division. For example, in $9:30_t \sim 9:30_{t+1}$ time division on January 5th, $T_{weighted}$ from January 5th, 9:30 to January 6th, 9:30 are summed up to generate $TO_t$ for January 5th.

In addition, calendar anomalies (holiday effect, day-of-the-week effect,) are common in financial markets (*Berument & Kiymaz, 2001*; *Jacobs & Levy, 1988*; *Kiymaz & Berument, 2003*). The holiday effect, which leads to abnormal stock price fluctuation on Monday, has been extensively studied in the stock area (*Bujari, Furini & Laina, 2017*; *Berument & Kiymaz, 2001*; *Shiller, 2003*). The news released on weekends or holidays is one reason that changes investors' behavior (*Qiu, Song & Chen, 2022*). Inspired by *Qiu, Song & Chen (2022)*, we generate an equation to consider holiday effect as shown below (n is days of vacation):

$$TO_{modified-t} = e^{-n}TO_{t-n} + \ldots + e^{-1}TO_{t-1} + TO_t \tag{7}$$

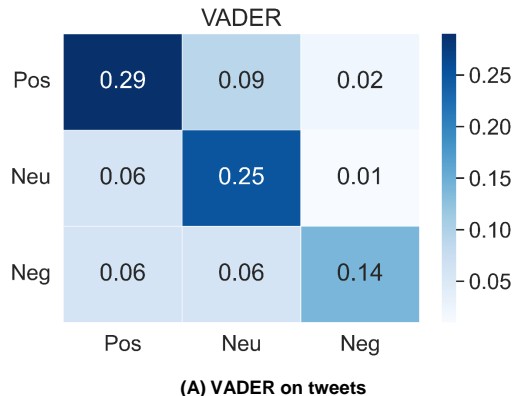

**(A) VADER on tweets**

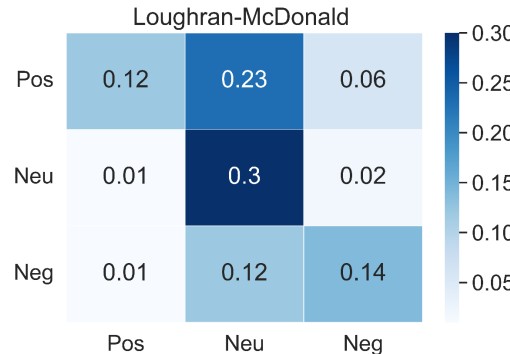

**(B) Loughran-McDonald dictionary on tweets**

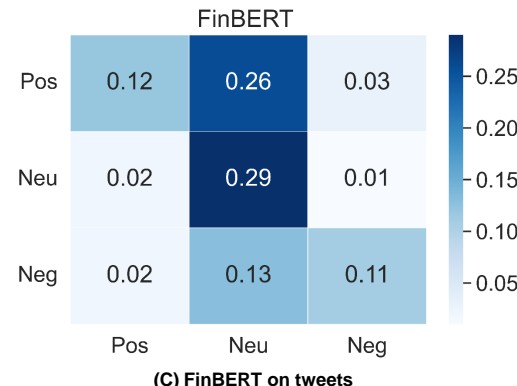

**(C) FinBERT on tweets**

**Figure 2** VADER, Loughran-McDonald dictionary and FinBERT performance on Tweets_labelled_09042020_16072020.

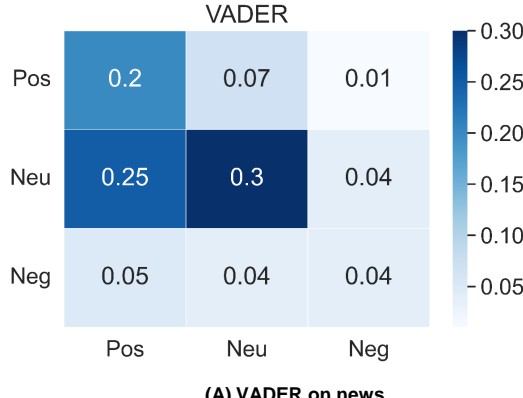

**(A) VADER on news**

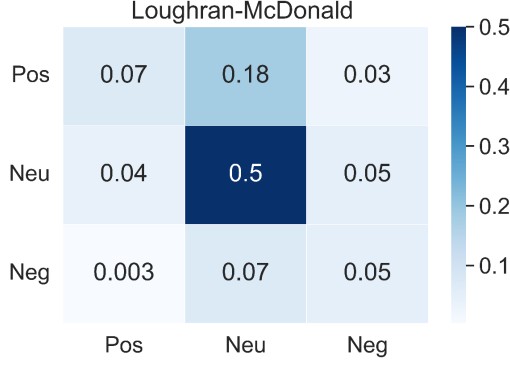

**(B) Loughran-McDonald dictionary on news**

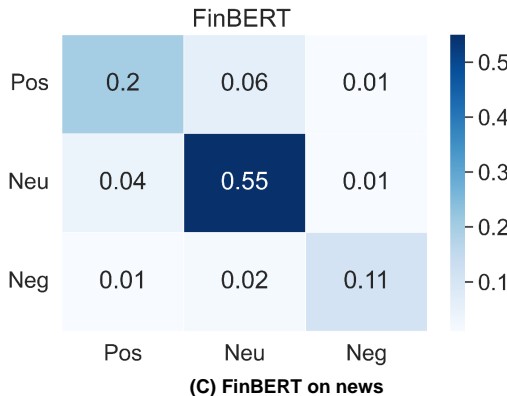

**(C) FinBERT on news**

**Figure 3** VADER, Loughran-McDonald dictionary and FinBERT performance on financial Phrase-Bank.

$$TN_{modified-t} = e^{-n}TN_{t-n} + \ldots + e^{-1}TN_{t-1} + TN_t. \tag{8}$$

Modified score $NO_{modified-t}$ and $NN_{modified-t}$ are also similar to get, just to change Tweets value to corresponding News value.

Take $TN_{modified-t}$ as an example, if $n = 2$ days, the $TN_{t-2}$, $TN_{t-1}$ and $TN_t$ can stand for summed $T_{weighted}$ on t = Sunday, t-1 = Saturday, t-2 = Friday. As sentiments have a stronger impact more recently, sentiment in the past decreasing exponentially (*Barberis et al., 2015*). Therefore, we concentrate weekend sentiments on Sunday and use modified Sunday weighted score to predict Monday's trend.

Finally, we will combine modified news and tweets score according to function Combine Natural Hours *CN* and Combine Opening Hours *CO*:

$$CN = \alpha TN_{modified-t} + (1 - \alpha) NN_{modified-t} \tag{9}$$

$$CO = \alpha TO_{modified-t} + (1 - \alpha) NO_{modified-t} \#(10). \tag{10}$$

In our experiment, result is the best when $\alpha = 0.25$. Thus, we apply 0.25 in the function to get *CO* and *CN*.

## Machine learning

*CO, CN* and *GoalI*, *GoalII* are aligned in time series. Sentiment features are the input in KNN, Tree (*scikit-learn, 2023*; *Biau et al., 2018*), SVM, random forest (RF), Naïve Bayes (NB), logistic regression (LR) algorithms with 10-fold cross validation to check the *GoalI*, *GoalII* prediction accuracy separately. This step was implemented by using Spyder and Orange from Anaconda. The parameters of the machine learning models are shown below:

- Random Forest: n_estimators =10, max_depth =None, min_samples_split =5.
- Naïve Bayes: MultinominalNB, alpha =1.0, force_alpha =False, fit_prior =True, class_prior =None.
- Logistic Regression: Penalty =L2 (Ridge), $C = 1$.
- KNN: n_neighbors =5, metric ='cosine'
- SVM: Kernel =Linear, Cost = 1, Regression loss epsilon = 0.1,
- Tree: Splitter = best, min_samples_leaf = 2, min_samples_split = 5, max_depth =100.

## Evaluation methodology

To evaluate the performance of different machine learning algorithms, we employ Classification Accuracy (CA), Precision, Recall, and F1-score. Equations of these four are shown in the following:

$$Accuracy = \frac{TP + TN}{TP + TN + FP + FN} \tag{11}$$

$$Precision = \frac{TP}{TP + FP} \tag{12}$$

$$Recall = \frac{TP}{TP + FN} \tag{13}$$

$$F1 - score = \frac{2\,Precision * Recall}{Precision + Recall} \qquad (14)$$

TP stands for true positive classification, and FP is also positive classification but false. TN stands for true negative classification, and FN is the positive result but wrongly classified as negative.

## RESULTS

Tables 1∼4 presents the six algorithms' performance for AAPL, AMZN, MSFT, NFLX in *Goal I: Open$_{t+1}$- Open$_t$* and Tables 5∼8 are the performance for those companies in *Goal II: Open$_{t+1}$- Close$_t$*. As the result shows, Naïve Bayes is the best classification algorithm in our model, six out of eight best classification accuracy comes from Naïve Bayes.

In *Goal I: Open$_{t+1}$- Open$_t$* prediction, all best Accuracy, F1-score, Precision, and Recall, except AAPL's F1-Score, are generated by using *CN*. In contrast, all best results in *Goal II: Open$_{t+1}$- Close$_t$* prediction are achieved with *CO*.

Therefore, since Naive Bayes algorithm has proven to be the most effective, it demonstrates strong performance in TD1 for Goal I (*Open$_{t+1}$- Open$_t$*) and even better performance in TD2 for Goal II (*Open$_{t+1}$ - Close$_t$*). To determine the overall performance, we calculate the average of Naive Bayes accuracy from both Goal I in TD1 and Goal II in TD2.

**TD1: Natural hours' time division from 0:00$_t$∼0:00$_{t+1}$**
**TD2: Opening hours' time division from 9:30$_t$∼9:30$_{t+1}$**
**Bold values represent the best performances**

## CONCLUSION

In this article, we develop a new weighted sentiment index $T_{weighted}$ by using Twitter retweets counts and VADER sentiment score, and we utilize FinBERT to generate $N_{weighted}$ to represent news opinion value. We propose a new time division, opening hours division, to study how tweets and news released in different time periods can predict next-day stock movement. Based on that, the summation of scores in different time divisions is calculated. In addition, the holiday effect are also considered in this article to construct modified score, which proved to be a more reliable and realistic indicator (*Qiu, Song & Chen, 2022*). Finally, we apply KNN, Tree, SVM, random forest, Naïve Bayes, logistic regression algorithms on aligned series to evaluate the performance of combined score *CN* and *CO* in predicting *Goal I: Open$_{t+1}$ - Open$_t$* and *Goal II: Open$_{t+1}$ - Close$_t$*.

The experimental result shows that Naïve Bayes is the best classification algorithm among six, six out of eight best results are produced by Naïve Bayes. The possible reason is that the dataset might be highly skewed, with one class being much more prevalent than the others. NB is known to perform well in such scenarios. Furthermore, *CN* is better in predicting *Goal I: Open$_{t+1}$ - Open$_t$*. Except for AAPL's F1-score, all best Accuracy, F1-score, Precision and Recall are generated by using *CN*. In reverse, *CO* presents all best results in *Goal II: Open$_{t+1}$ - Close$_t$* prediction. It proves that non-natural time division

**Table 1  Performance of six ML classifiers on AAPL stock's Goal I.**

| Model | Accuracy | | F1−score | | Precision | | Recall | |
|---|---|---|---|---|---|---|---|---|
| | TD1 | TD2 | TD1 | TD2 | TD1 | TD2 | TD1 | TD2 |
| KNN | 0.532 | 0.569 | 0.528 | 0.569 | 0.529 | 0.569 | 0.532 | 0.569 |
| LR | 0.600 | 0.502 | 0.551 | 0.351 | **0.635** | 0.270 | 0.600 | 0.502 |
| NB | **0.604** | 0.545 | 0.577 | 0.543 | 0.616 | 0.544 | **0.604** | 0.545 |
| RF | 0.552 | 0.584 | 0.551 | **0.584** | 0.551 | 0.584 | 0.552 | 0.584 |
| SVM | 0.484 | 0.471 | 0.373 | 0.304 | 0.547 | 0.224 | 0.484 | 0.471 |
| Tree | 0.496 | 0.541 | 0.488 | 0.538 | 0.489 | 0.539 | 0.496 | 0.541 |

Bold values represent the best performances.

**Table 2  Performance of six ML classifiers on AMZN stock Goal I.**

| Model | Accuracy | | F1−score | | Precision | | Recall | |
|---|---|---|---|---|---|---|---|---|
| | TD1 | TD2 | TD1 | TD2 | TD1 | TD2 | TD1 | TD2 |
| KNN | 0.600 | 0.514 | 0.597 | 0.510 | 0.597 | 0.509 | 0.600 | 0.514 |
| LR | 0.600 | 0.545 | 0.582 | 0.385 | 0.599 | 0.297 | 0.600 | 0.545 |
| NB | **0.624** | 0.510 | **0.600** | 0.421 | **0.633** | 0.446 | **0.624** | 0.510 |
| RF | 0.540 | 0.561 | 0.540 | 0.561 | 0.540 | 0.561 | 0.540 | 0.561 |
| SVM | 0.476 | 0.475 | 0.428 | 0.387 | 0.507 | 0.529 | 0.476 | 0.475 |
| Tree | 0.540 | 0.510 | 0.537 | 0.505 | 0.536 | 0.505 | 0.540 | 0.510 |

Bold values represent the best performances.

**Table 3  Performance of six ML classifiers on MSFT stock Goal I.**

| Model | Accuracy | | F1−score | | Precision | | Recall | |
|---|---|---|---|---|---|---|---|---|
| | TD1 | TD2 | TD1 | TD2 | TD1 | TD2 | TD1 | TD2 |
| KNN | 0.520 | 0.451 | 0.515 | 0.443 | 0.515 | 0.441 | 0.520 | 0.451 |
| LR | 0.536 | 0.533 | 0.380 | 0.383 | 0.294 | 0.406 | 0.536 | 0.533 |
| NB | **0.600** | 0.482 | **0.573** | 0.417 | **0.604** | 0.428 | **0.600** | 0.482 |
| RF | 0.464 | 0.463 | 0.459 | 0.459 | 0.457 | 0.458 | 0.464 | 0.463 |
| SVM | 0.488 | 0.502 | 0.417 | 0.433 | 0.546 | 0.570 | 0.488 | 0.502 |
| Tree | 0.472 | 0.494 | 0.467 | 0.484 | 0.465 | 0.484 | 0.472 | 0.494 |

Bold values represent the best performances.

is particularly useful in predicting specific goals. Based on this result, we proposed our model for predicting *Goal I: Open$_{t+1}$ - Open$_t$* and *Goal II: Open$_{t+1}$ - Close$_t$* separately in Figs. 4 and 5. Compared with Kabbani and Usta's result (*Kabbani & Usta, 2022*), which gets 63.6% best accuracy by using seven features (including sentiment score and technical factors). Our model gets the best 62.4% accuracy with only news and tweets sentiment features.

**Table 4  Performance of six ML classifiers on NFLX stock Goal I.**

| Model | Accuracy | | F1−score | | Precision | | Recall | |
|---|---|---|---|---|---|---|---|---|
| | TD1 | TD2 | TD1 | TD2 | TD1 | TD2 | TD1 | TD2 |
| KNN | 0.500 | 0.443 | 0.498 | 0.442 | 0.499 | 0.442 | 0.500 | 0.443 |
| LR | **0.588** | 0.494 | **0.588** | 0.364 | **0.589** | 0.409 | **0.588** | 0.494 |
| NB | 0.512 | 0.478 | 0.507 | 0.468 | 0.511 | 0.473 | 0.512 | 0.478 |
| RF | 0.468 | 0.478 | 0.468 | 0.478 | 0.468 | 0.480 | 0.468 | 0.478 |
| SVM | 0.496 | 0.502 | 0.409 | 0.479 | 0.505 | 0.510 | 0.496 | 0.502 |
| Tree | 0.512 | 0.502 | 0.508 | 0.502 | 0.511 | 0.502 | 0.512 | 0.502 |

Bold values represent the best performances.

**Table 5  Performance of six ML classifiers on AAPL stock Goal II.**

| Model | Accuracy | | F1−score | | Precision | | Recall | |
|---|---|---|---|---|---|---|---|---|
| | TD1 | TD2 | TD1 | TD2 | TD1 | TD2 | TD1 | TD2 |
| KNN | 0.488 | 0.570 | 0.482 | 0.569 | 0.480 | 0.567 | 0.488 | 0.570 |
| LR | 0.568 | 0.570 | 0.412 | 0.417 | 0.323 | 0.329 | 0.568 | 0.570 |
| NB | 0.524 | **0.609** | 0.419 | **0.585** | 0.415 | **0.600** | 0.524 | **0.609** |
| RF | 0.540 | 0.516 | 0.539 | 0.515 | 0.537 | 0.514 | 0.540 | 0.516 |
| SVM | 0.532 | 0.484 | 0.464 | 0.476 | 0.481 | 0.518 | 0.532 | 0.484 |
| Tree | 0.544 | 0.539 | 0.543 | 0.537 | 0.542 | 0.536 | 0.544 | 0.539 |

Bold values represent the best performances.

**Table 6  Performance of six ML classifiers on AMZN stock Goal II.**

| Model | Accuracy | | F1−score | | Precision | | Recall | |
|---|---|---|---|---|---|---|---|---|
| | TD1 | TD2 | TD1 | TD2 | TD1 | TD2 | TD1 | TD2 |
| KNN | 0.532 | 0.551 | 0.524 | 0.542 | 0.520 | 0.537 | 0.532 | 0.551 |
| LR | 0.616 | **0.621** | 0.470 | 0.476 | 0.379 | 0.386 | 0.616 | **0.621** |
| NB | 0.616 | **0.621** | 0.470 | 0.476 | 0.379 | 0.386 | 0.616 | **0.621** |
| RF | 0.488 | 0.520 | 0.486 | 0.518 | 0.484 | 0.517 | 0.488 | 0.520 |
| SVM | 0.604 | 0.547 | 0.471 | 0.533 | 0.454 | 0.526 | 0.604 | 0.547 |
| Tree | 0.488 | 0.594 | 0.483 | **0.582** | 0.480 | **0.578** | 0.488 | 0.594 |

Bold values represent the best performances.

## Funding

The authors received no funding for this work.

## Competing Interests

The authors declare there are no competing interests.

**Table 7  Performance of six ML classifiers on MSFT stock Goal II.**

| Model | Accuracy | | F1−score | | Precision | | Recall | |
|---|---|---|---|---|---|---|---|---|
| | TD1 | TD2 | TD1 | TD2 | TD1 | TD2 | TD1 | TD2 |
| KNN | 0.492 | **0.590** | 0.488 | **0.587** | 0.485 | **0.586** | 0.492 | **0.590** |
| LR | 0.560 | 0.578 | 0.411 | 0.424 | 0.324 | 0.334 | 0.560 | 0.578 |
| NB | 0.560 | 0.551 | 0.430 | 0.501 | 0.468 | 0.514 | 0.560 | 0.551 |
| RF | 0.496 | 0.512 | 0.493 | 0.513 | 0.491 | 0.516 | 0.496 | 0.512 |
| SVM | 0.444 | 0.508 | 0.397 | 0.506 | 0.487 | 0.504 | 0.444 | 0.508 |
| Tree | 0.528 | 0.523 | 0.515 | 0.514 | 0.513 | 0.511 | 0.528 | 0.523 |

Bold values represent the best performances.

**Table 8  Performance of six ML classifiers on NFLX stock Goal II.**

| Model | Accuracy | | F1−score | | Precision | | Recall | |
|---|---|---|---|---|---|---|---|---|
| | TD1 | TD2 | TD1 | TD2 | TD1 | TD2 | TD1 | TD2 |
| KNN | 0.516 | 0.555 | 0.514 | 0.547 | 0.512 | 0.546 | 0.516 | 0.555 |
| LR | 0.556 | 0.563 | 0.403 | 0.408 | 0.316 | 0.320 | 0.556 | 0.563 |
| NB | 0.560 | **0.594** | 0.459 | **0.568** | 0.526 | **0.584** | 0.560 | **0.594** |
| RF | 0.488 | 0.566 | 0.489 | 0.566 | 0.491 | 0.566 | 0.488 | 0.566 |
| SVM | 0.484 | 0.531 | 0.442 | 0.482 | 0.540 | 0.494 | 0.484 | 0.531 |
| Tree | 0.492 | 0.551 | 0.486 | 0.546 | 0.484 | 0.544 | 0.492 | 0.551 |

Bold values represent the best performances.

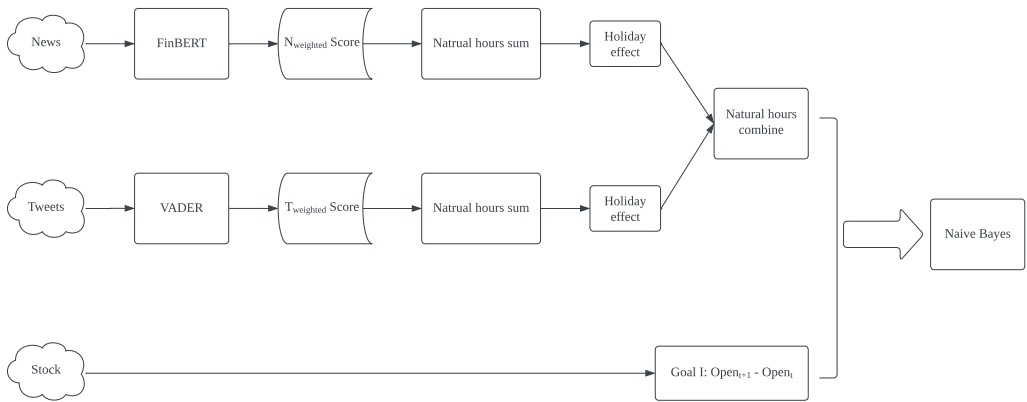

**Figure 4  Proposed model for predicting *Goal I: Open_{t+1}-Open_t*.**

## Author Contributions

- Qianyi Xiao conceived and designed the experiments, performed the experiments, analyzed the data, performed the computation work, prepared figures and/or tables, authored or reviewed drafts of the article, and approved the final draft.
- Baha Ihnaini conceived and designed the experiments, prepared figures and/or tables, authored or reviewed drafts of the article, and approved the final draft.

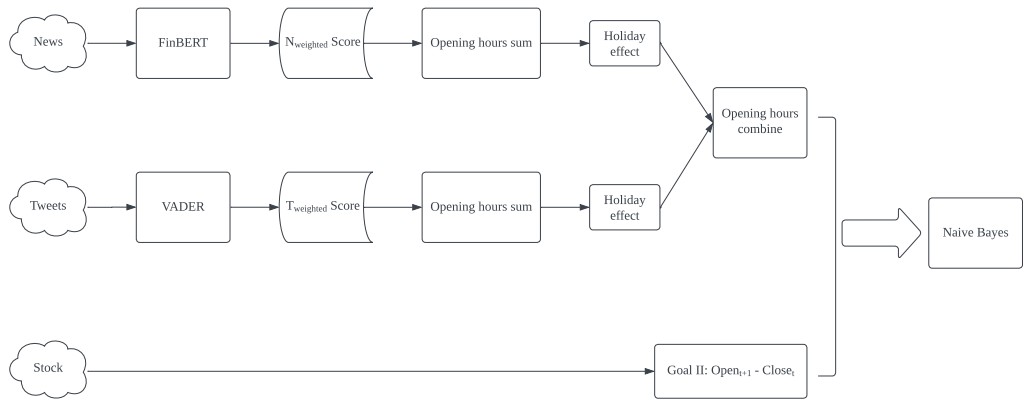

**Figure 5** Proposed model for predicting *Goal II: Open$_{t+1}$ - Close$_t$.*

## Data Availability

The raw data is available in the Supplemental Files.

## Supplemental Information

Supplemental information for this article can be found online at http://dx.doi.org/10.7717/peerj-cs.1293#supplemental-information.

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
