# Peer review of "Stock trend prediction using sentiment analysis"

_PeerJ Computer Science, doi:10.7717/peerj-cs.1293_

## Round 0.1 · original submission · Major Revisions

The paper needs significant modifications.

Reviewer 1 ·

Basic reporting

1. The manuscript is well-written, with no typos except for Tables 1 and 2. They do not appear. You may need to re-label the tables.

Experimental design

2. If the class label (Goal 1 or 2) is based on a one-dimensional scalar input value (i.e., CN or CO), why are ML models used? Isn't it better to use a threshold or compute the correlation to show the effectiveness of CN and CO scores?
3. What is the "Tree" classifier? Can you cite it?
4. What are the parameters of the ML models?
5. How are the ML models implemented?
6. Since training is applied during cross validation, I wonder why FinBERT is not finetuned using training data and Transfer learning techniques.
7. Fig.2 and 3, which show the relationship between NB accuracy and volume, could be combined in one stacked bar chart.
8. Could you please explain why VADER and FinBERT were chosen but not the Loughran-McDonald dictionary in the Sentiment Analysis Test Result Subsection?

Validity of the findings

9. Can you please explain why the opening hours division is better than the natural hours division?
10. Can you show how the experiments concluded that the opening hours division is better than the natural hours division?
11. The authors may have to show the performance of other related approaches on the same testing dataset.
12. Can you please show the performance of TN, TO, NN, and NO separately?
13. Can you explain why NB got the maximum accuracy?
14. Why RandomForest got the highest F1?
15. The authors may have to balance the four stock datasets using data augmentation or resampling, and show the results.

Reviewer 2 ·

Basic reporting

The article is fluent and readable.
The background section seems complete, although I think it is important to include alternative references such as (https://www.mdpi.com/2078-2489/13/11/524). These papers focus purely on numerical data, but it is important to cite them.
T_weighted, N_weighted, and the other notations, such as Goal_I, are very well explained.

Experimental design

The primary motivation that led the authors to propose the measures that are the heart of the paper lies in the fact that most previous research adds up the text sentiment score on each natural day and uses that aggregate score for that aggregate score to predict various headline trends.
The underlying design seems reasonable, and it all seems to work well.
The research questions seem to be well and accurately asked.
However, there has not been a rigorous investigation. First of all, the figures you use should have a better resolution.
Second, vader is a bad and old model, especially in recent stock and stock sentiment.
It would help if you did an ablation study by removing portions of your architecture.

Validity of the findings

The ablation study will be the key to your research question. Despite the various metrics you put out, there is something that needs to sound right scientifically. An ML model outperforms the performance of FinBERT. This fact seems very strange.

Additional comments

Indeed, the correlation between data volume and accuracy is correct, but I do not think it is some hot water discovery.

Reviewer 3 ·

Basic reporting

1. Clear and unambiguous, professional English used throughout.
2. Literature references, sufficient field background/context provided.
3. Professional article structure, figures, tables. Raw data shared.

Experimental design

1.Original primary research within Aims and Scope of the journal.
2.Methods described with sufficient detail & information to replicate.

Validity of the findings

1.All underlying data have been provided; they are robust, statistically sound, & controlled.
2.Conclusions are well stated, linked to original research question & limited to supporting results.

---

## Round 0.2 · accepted · Accept

I confirm that the authors have addressed all of the reviewers' comments and the manuscript is ready for publication.

Reviewer 1 ·

Basic reporting

Clear and professional English used throughout.
Literature references, sufficient field background/context provided.

Experimental design

Original primary research within the Aims and Scope of the journal.

Validity of the findings

Conclusions are well stated, linked to original research questions & limited to supporting results.

Additional comments

I appreciate the authors' effort to address my comments.

Reviewer 2 ·

Basic reporting

The manuscript is clear and understandable. The authors introduce the topic very well.
The literature and the references provided are sufficent.

Experimental design

The line of research is clear and understandable. This line cover the aims and scope of the journal.

Validity of the findings

The results are demonstrated and justified well. I indicate an ablatio study to understand your work's weak/strong components.